# Inter-Set Foam Rolling of the Latissimus Dorsi Acutely Increases Repetitions in Lat Pull-Down Exercise without Affecting RPE

**DOI:** 10.3390/jfmk9010043

**Published:** 2024-02-29

**Authors:** Luca Russo, Sebastiano Riccio, Giulio Zecca, Alin Larion, Riccardo Di Giminiani, Johnny Padulo, Cristian Popa, Gian Mario Migliaccio

**Affiliations:** 1Department of Human Sciences, Università Telematica degli Studi IUL, 50122 Florence, Italy; l.russo@iuline.it (L.R.); g.zecca@iuline.it (G.Z.); 2Department of Medicine and Health Sciences, University of Molise, 86100 Campobasso, Italy; ricciooosebb@gmail.com; 3Faculty of Physical Education and Sports, Ovidius University of Constanta, 900029 Constanta, Romania; alinlarion@yahoo.com (A.L.); crispopa2002@yahoo.com (C.P.); 4Department of Biotechnological and Applied Clinical Sciences, University of L’Aquila, 67100 L’Aquila, Italy; riccardo.digiminiani@univaq.it; 5Department of Biomedical Sciences for Health, Università degli Studi di Milano, 20133 Milan, Italy; 6Department of Human Sciences and Promotion of the Quality of Life, San Raffaele Rome Open University, 00166 Rome, Italy; gianmario.migliaccio@uniroma5.it

**Keywords:** foam roller, self-myofascial release, training, upper limb, prevention

## Abstract

Foam rolling is widely used in fitness, sports, rehabilitation, and injury prevention. However, there are limited data available on the effect of foam rolling techniques on the upper limbs. The aim of this investigation is to assess the effects of foam rolling the latissimus dorsi area during the rest period between two consecutive lat pull-down exercise (LPDE) sets. Seventeen resistance training experienced volunteer male subjects (25.8 ± 3.4 years; 180.3 ± 9.0 cm; 79.7 ± 9.9 kg) participated in this research. Each subject performed 2 training sessions of LPDE in a random order, separated by one week. Each session consisted of 2 sets of maximum repetitions using 85% of their one-repetition maximum (1RM), with a 7 min rest period between sets. The rest period condition between sets was different in the 2 sessions: passive rest (Pr) or foam rolling the latissimus dorsi muscle bilaterally for 3 sets of 45 s (FRr). The following variables were assessed for each LPDE set: number of repetitions (REPS), average excursion per repetition in millimeters (EXC), average power of the set in watts (AP) and rating of perceived exertion (RPE). Pr did not show any significant change between the first and the second LPDE set for REPS, EXC, and AP. However, there was a significant increase for RPE (8.4 ± 0.5 vs. 8.9 ± 0.5 a.u., *p* = 0.003) between the two sets. FRr resulted in an increase for REPS (7.1 ± 1.5 vs. 8.2 ± 1.3, *p* < 0.001) and AP (304.6 ± 61.5 W vs. 318.8 ± 60.8 W, *p =* 0.034) between the first and the second LPDE sets, but no changes were observed for EXC and RPE. The use of foam rolling techniques on the latissimus dorsi area during the complete rest period between sets in LPDE at 85% 1RM appears to improve the number of repetitions and the movement power without affecting the RPE during the second set.

## 1. Introduction

The foam roller, a versatile tool used by health professionals, personal trainers, strength and conditioning coaches, and physical therapists, is primarily used for its effectiveness in myofascial release [1,2,3], improving mobility, and relieving muscle soreness [4]. Its ease of use makes it ideal for both warm-up routines and post-exercise recovery [5], thereby promoting muscular health and contributing to overall well-being [6].

Foam rolling exercises can affect the myofascial tissues, as well as massaging yourself on specific painful points using other various tools (i.e., hard balls or plastic sticks) [7].

Scientific evidence broadly supports the use of foam roller as a beneficial adjunct in sports performance, as well as in health promotion and maintenance training. Foam rollers can be vibrating or non-vibrating [8]. Regardless of their technical characteristics, research has underscored a series of advantages in performing a foam roller self-massage before the training session, such as a significant reduction in muscle fatigue and a temporary increase in range of motion without compromising muscle strength [9,10], as well as the attenuation of delayed onset muscle soreness, promoting quicker functional recovery [11,12].

Nonetheless, variability in study methodologies and the diversity in research protocols highlight the need for further investigation to refine the application of this approach. For example, despite the burgeoning interest in foam rolling, there is a discernible dearth of research focused on the upper extremities. Indeed, the majority of scientific information about the effects of foam rolling is available for the lower extremities. Quadriceps, hamstring, gluteal muscles, calves, and intrinsic foot muscles are the main targets of foam rolling studies [6,13,14,15]. The upper limbs may be less suitable for foam rolling applications, likely due to the limited contact surface the upper limb muscles can provide to the foam roller. Indeed, it is not coincidental that only a limited number of studies have explored the effects of foam rolling on the upper limbs, making this study a novelty. Given this practice, acquiring scientific insights into the effects of foam rolling on upper limb muscles can prove highly valuable for exercise professionals. Two major muscles that can benefit from foam roller exercises are the pectoralis major [16] and latissimus dorsi [17]. The latter is an adductor and internal rotator of the humerus. It is a sizable muscle and it is very easy to treat with foam roller due to its anatomical pathway. The latissimus dorsi is commonly targeted in the gym through the lat pull-down exercise (LPDE), a standard strength training exercise performed using a dedicated isotonic machine, typically referred to as a Lat-Machine. This exercise involves sitting and pulling down a bar, initially with both arms extended upward. The downward pulling motion occurs along the frontal plane, facilitating the activation of the latissimus dorsi to adduct the arms [18].

To the best of our knowledge, no scientific research has previously been conducted on foam rolling of the latissimus dorsi and its effects on LPDE performance. Given the widespread use of foam rolling and LPDE in the gyms, it can be crucial to gather data on their possible interaction. The effects of pre-training foam rolling on performance are well-documented in the literature [5,9,10,11,12], yet limited data exist regarding the effects of inter-set foam rolling. Two studies by Monteiro et al. [19,20] suggest that incorporating foam rolling between sets of knee extension exercises may decrease the subsequent number of repetitions to exhaustion; regardless of whether foam rolling is applied to agonistic or antagonistic muscles. However, to our knowledge, no similar experimental data have been published on upper limbs or multi-joint exercises such as LPDE. For these reasons, this pilot study aimed to investigate the effects of foam rolling the latissimus dorsi area during the rest period between two consecutive sets of LPDE. Building upon the findings of previous literature [19,20], the hypothesis of this investigation was that the use of inter-set foam rolling procedures could negatively impact the LPDE performance in the subsequent set.

## 2. Materials and Methods

### 2.1. Participants

Trained and expert people in resistance training were needed for this research. Twenty-five male volunteer individuals were recruited for this study, but only seventeen of them (age 25.8 ± 3.4 years; body height 180.3 ± 9.0 cm; body mass 79.7 ± 9.9 kg) met the following inclusion criteria: (1) regular resistance training for at least 2 years; (2) regular performance of LPDE once a week at least; (3) no shoulder pain; (4) no muscular-skeletal pathologies for the upper limbs. The participants were recruited within a fitness gym center located in Campobasso (Italy). The sample size was checked using G*Power software version 3.1.9.7 and setting the error probability at 5% and the effect size at 0.7; according to these data, seventeen subjects were sufficient to guarantee a power analysis higher than 80%. The protocol conformed to internationally accepted policy statements regarding the use of human participants in accordance with the Declaration of Helsinki Declaration and was approved by the Ovidius University of Constanta Nr. 263 11 April 2023. All participants gave their written informed consent to participate in the study after receiving a thorough explanation of the study’s protocol. No risks were forecasted for the participants and the testing environment was controlled.

### 2.2. Instrumentations and Measurements

The self-massage on the latissimus dorsi area was performed using a foam roller consisting on a heavy-duty plastic cylinder covered by a high-density foam layer (ATS, Arezzo, Italy—dimensions 0.34 × 0.14 m; weight 0.454 kg).

The LPDE (Figure 1) was assessed using a linear encoder (Chronojump-Boscosystem, Barcelona, Spain) hooked to the weight stack, aiming to measure the kinematics of the motion. The reliability and concurrent validity of the linear encoder was previously assessed [21] and standard calibration procedures were performed before each test. For each test, the linear encoder was used to measure: the number of repetitions (REPS), the average excursion in millimeters per repetition (EXC), the average power in watts per set (AP). It is worth noting that the EXC is a fundamental parameter for this kind of study because it represents the total excursion of the weight stack form the starting arms position to the end of the repetition. A common mistake when performing LPDE is the reduction of ROM due to fatigue during repetitions; therefore, the measurement of the EXC is a very practical and easy way to monitor the technical execution consistency.

Two scales were used to assess separately the LPDE induced fatigue and the self-massage induced pain/discomfort. For the fatigue, immediately after each LPDE set, the participants indicated their rating of perceived exertion using the category rating-10 (CR-10) scale modified by Foster et al. [22]. For the pain/discomfort, immediately after the foam rolling session, the participants indicated their perceived pain using a 0–10 Numerical Rating Scale (NRS) [7,14,23].

### 2.3. Procedure and Data Collection

This research has a cross-sectional design with a random crossover approach. Training and testing procedures were carried out in a fitness gym center at a mean temperature of 20 °C and a mean relative humidity of 51%, and each subject was tested at the same time of the day, avoiding any circadian effect [24,25,26]. The timeframe of the study engaged each participant for six weeks (Figure 2). In the following paragraphs, a detailed description of each phase is provided.

#### 2.3.1. Familiarization Procedures

Familiarization procedures were performed for two weeks (W1–W2 in Figure 2) before testing started. During this period, a kinesiologist [27] highly skilled in resistance training (S.R.) explained the proper technique to perform the LPDE for a total amount of six training sessions (three sessions per week). The aim of these familiarization procedures was the correction of some individual technical errors using a normalized grip width on the bar for all participants. The grip width was selected by measuring and doubling the biacromial length, as suggested by the literature [28]. The individual grip width for each participant was registered by the kinesiologist and was used during each LPDE session during the whole length of the study. It is worth remembering that all the participants were experts in resistance training, therefore six familiarization sessions were considered sufficient to correct some technical details. At the end of the familiarization procedure, the LPDE was not performed for a week (W3 in Figure 2) and after this period, the testing procedure began. 

#### 2.3.2. Tests

The testing procedure lasted three weeks for each subject (W4-W5-W6 in Figure 2); a total of three days of test separated by one week of rest were performed. During that resting period between two consecutive days of test, the participants were allowed to train regularly in the gym up to, and no further than, two days before the test day. Participants had only one limitation on physical training along the experimental period: they could not perform any specific exercise with target on the latissimus dorsi muscle (i.e., rowing or other vertical pulling exercises) except the LPDE in the days of test. This choice was mandatory to avoid any kind of influence on the LPDE performance during the tests. In the following paragraphs, more details are provided on the test period.

#### 2.3.3. First Week of Test—1 RM Measurement

The first week of tests (W4 in Figure 2) was used to perform the preliminary test phase. Explanation of the procedures, participants’ anthropometric data acquisition, and measurements of the 1RM for LPDE were performed. Due to the participants’ nature (experienced gym attenders), the 1RM measurement was performed using the Brzycki protocol [29], an indirect method very popular among resistance training [30].

Before the procedure for the 1 RM calculation each participant performed the same general and specific warm-up (Table 1).

Immediately after the warm-up, each participant performed a gradual load ramping on 5 repetitions with an incremental recovery ranging from 2 to 3 min based on the load increment. The same skilled kinesiologist followed each participant during the procedure, aiming to find out the maximum load with whom performing maximum 5 technically correct repetitions. That load was used to calculate the theoretical 1RM using the following formula [29]:$$1RM=\frac{Load}{1.0278-(0.0278\times Reps)}$$

“Load” is the weight in kg used to perform maximum 5 technically correct repetitions and “Reps” is the number of repetitions (5).

The valued 1 RM was used to establish the individual training load during the next days of test, in which all the participants were tested using the 85% of the valued 1RM.

#### 2.3.4. Second and Third Week of Test—LPDE Trials and Foam Rolling Training

The last two weeks of tests (W5–W6 in Figure 2) were used to assess the acute effects of using a foam roller during the resting period between two sets of LPDE at maximum repetitions performed with 85% of 1RM. Each participant, in random order, performed one week apart two training sessions, one session with and one without foam rolling intervention.

In both sessions, each participant performed the same general and specific warm-up (Table 2).

Immediately after the warm-up procedure, the LPDE training sessions began. The two sessions were identical, only the rest modalities between the two LPDE sets were different.

Two sets of LPDE were performed at 85% 1 RM for maximum repetitions. The same kinesiologist monitored the exercise for each training session. The LPDE was stopped in two cases: (1) muscular exhaustion; or (2) when the kinesiologist assessed a fatigue induced technical fail for two consecutive repetitions. For each set, the following parameters were collected: (1) number of repetitions (REPS); average excursion per repetition in millimetres (EXC); average power of the set in watts (AP); and the rating of perceived exertion (RPE). In particular, the AP measurement has been performed with the aim of monitoring and comparing the speed of the two LPDE sets in both recovery conditions. To avoid any kind of psychological influence induced by the load, the kinesiologist hid the weight stack during each set for each participant. Even the music speakers were shut down in that part of the gym during the testing procedures, aiming to avoid differences in acoustic stimuli between participants.

The foam rolling recovery (FRr) lasted 7 min and consisted of: (1) 2′30″ of passive recovery and then; (2) 3 sets of 45″ of latissimus dorsi area bilateral foam rolling (total foam rolling time 4′30″). At the end of foam rolling procedures the self-perceived pain was assessed using a 0–10 NRS. The self-perceived pain averaged at 5.9 ± 2.1 A.U. for all participants. Foam rolling between sets was performed lying on the ground laterally with the foam roller positioned under the latissimus dorsi muscle skin area, between the proximal third of the humerus and the distal part of the scapula (Figure 3). Three bouts of myofascial release self-massage were performed for each side, each bout lasted 45 s. The kinesiologist randomly assigned the starting side, and the work sequence was switched back and forth between the left and right side. The adopted foam rolling techniques were adherent to other literature protocols [14]. In the first bout, the trigger point was identified and pressed (Figure 3a). In the second bout, slow sliding and transversal circling movements were performed around the trigger point (Figure 3b). In the third and last bout, pressure was still applied to the trigger point combining abduction and back movements of the arm (Figure 3c).

On the other hand, the passive recovery (Pr) consisted of 7 min of complete recovery between the two LPDE sets.

The two recovery conditions (FRr/Pr) were randomly administered through a simple draw performed immediately before the first training session. The kinesiologist registered the results of the draw.

### 2.4. Statistical Analysis

The normality of the data was assessed using the Shapiro–Wilk test. Although most of the data followed a normal distribution, non-parametric tests were applied for the analysis because the sample size. Differences at baseline between the two different experimental conditions were tested with a Mann–Whitney test. A Wilcoxon test was used to compare the two consecutive sets of LPDE for each recovery condition. The significance level was set at *p* 0.05, and the statistical analysis was performed using SPSS (SPSS Inc., Chicago, IL, USA).

## 3. Results

The LPDE was performed at 85% of 1RM, with the average load used by the sample being 75.5 ± 9.5 kg for each set in each recovery condition (Pr and FRr). Statistical analysis revealed no significant differences between the values of REPS, EXC, AP, and RPE (Table 3) for the first set performance in the two different test days both when a passive rest and a foam rolling rest were performed (Pr and FRr, respectively). That lack of significance confirms that both sessions started from the same baseline, and no training, learning, placebo, or psychological effects affected the baseline measurement.

Comparing the two LPDE sets performed with and without the foam roller during recovery (FRr and Pr, respectively), the statistical analysis revealed significant differences in REPS, AP, and RPE.

Specifically, FRr led to a significant increase in REPS (7.2 ± 1.5 vs. 8.2 ± 1.3 n in the first and second set, respectively, *p* = 0.003) and AP (304.6 ± 61.5 vs. 318.8 ± 60.8 n in the first and second set, respectively, *p* = 0.035) in the second set after recovery, with no change in perceived fatigue (RPE). On the other hand, Pr did not show any significant increase in any parameter except self-perceived fatigue; indeed, RPE increased in the second LPDE set in the Pr condition (8.4 ± 0.5 vs. 8.9 ± 0.5 n in the first and second set, respectively, *p* = 0.007). Table 4 contains all the results for FRr and Pr conditions, and Figure 4 and Figure 5 show the details of REPS and RPE changes between the first and second set.

## 4. Discussion

This investigation aimed to understand the effects of foam rolling on the latissimus dorsi area during the rest period between two consecutive sets of lat pull-down exercises (LPDE). The reason for using inter-set foam rolling lies in the fact that some individuals in the gym often stretch the agonist muscle between sets. Although inter-set stretching is scientifically investigated for its theoretical advantage in terms of muscular hypertrophy as a chronic adaptation [31,32,33,34], the acute effects are not well-investigated and appear to be detrimental for resistance training performance in the set subsequent to the stretching procedure [35,36]. On the other hand, the literature reports that pre-exercise foam rolling enhances the range of motion without compromising muscle strength [9,10], but it seems to negatively affect the subsequent performance when it is performed between sets in lower limbs exercise [19,20]. Therefore, it is rational to test the foam rolling inter-set effect in the upper limbs for confirming or denying the foam rolling advantages. Because no data are available about this topic for the latissimus dorsi muscle, the research hypothesis was that the use of foam rolling procedures between two consecutive sets of LPDE could negatively affect the performance of the subsequent set.

The results of the present investigation did not confirm the hypothesis, in contrast with previous investigations [19,20]. Not only did the use of the foam roller inter-set not affect the performance of the subsequent set, but it also appeared to improve performance in terms of repetitions and average power without increasing perceived fatigue. In contrast, passive rest significantly increased fatigue. A specific discussion of the results is needed to interpret the data properly and to design future lines of research. The effects of foam rolling on the latissimus dorsi muscle were previously investigated only in terms of range of motion improvements during complex movements [17]. Therefore, the current results are brand new in the research field of foam rolling, and this research can be considered the first to present these data on the latissimus dorsi muscle and LPDE.

It is worth noting that the significant improvement in REPS in the set after foam rolling is similar to a previous study that investigated the effects of antagonistic muscle stretching [37]. Discussing the improvement in repetitions during the set immediately after foam rolling is challenging in terms of neurophysiology, especially considering that a recent paper [38] showed how foam rolling procedures improve the range of motion similar to sham rolling. The paper’s conclusions attribute the local and non-local increase in range of motion to the warm-up effect of the movements performed to roll on the foam roller or on the sham board [39]. These results can be considered a game changer in foam rolling research. At the same time, in the current research, the range of motion during the LPDE was monitored using the EXC parameter. EXC represents the measurement of the average excursion per repetition in millimeters, and no changes in EXC were measured in either FRr or Pr, suggesting that the increase in REPS is not connected with an increase in range of motion. Moreover, another important aspect must be considered to interpret the results of the current investigation properly. The rest period between sets was very long (7 min), on one hand, to allow complete recovery after the first maximal exhaustion set, and on the other hand, to enable a correct stimulation of the latissimus dorsi using the foam roller. Considering this information, it could be rational to think that after such a long time, the Pr group started the second set of LPDE in a somewhat ‘deactivated’ manner. However, the data do not support this kind of argument. Indeed, although the RPE is significantly higher, suggesting an increase in fatigue, the REPS are the same between the first and second sets (REPS 7.1 ± 1.5 and 7.6 ± 1.3 n, respectively). The equality in REPS between the two sets for Pr confirms that the long rest between sets did not deactivate the subjects; on the contrary, the rest time allowed complete recovery, enabling the performance of another maximal exhaustion set of LPDE. Regarding the FRr group, the possibility of a warm-up inter-set effect of foam rolling procedures should be taken into consideration, although some doubts remain. Indeed, the FRr significantly increased one repetition between the first and second sets (REPS 7.2 ± 1.5 and 8.2 ± 1.3 n, respectively) and even AP (304.6 ± 61.5 and 318.8 ± 60.8 Watts, respectively). The improvement in AP suggests that FRr allowed participants to pull down the bar faster in the second set compared to the first one. This aspect cannot be explained only with Warneke et al.’s [38] suggestion of a warm-up effect during the rest period. It is necessary to consider the foam roller’s action on the soft tissues that probably allows an enhancement of the explosive parameters [39] and a better joint position sense of the upper limb [40]. Finally, it is crucial to underline the absence of an increase in RPE for FRr in the second set (8.4 ± 0.6 and 8.2 ± 0.7 n, respectively). The lack of modification in RPE after FRr, in contrast to the increase after Pr, is in line with previous literature [41].

The current investigation represents an absolute novelty in the literature on upper limb foam rolling, and several practical applications can be mentioned, such as: (1) its usefulness in reducing self-perceived fatigue in maximal exhaustion sets; (2) its effectiveness in maintaining the number of repetitions over the sets; (3) its potential in saving time during training by associating two different stimuli on the target muscle. These practical applications should be scientifically tested before having any certainty about their absolute effectiveness; therefore, more research is needed in this field.

### Limitations

A certain number of limitations need to be acknowledged to improve the quality of data through further investigations. This study demonstrates acute results only on two LPDE sets. Therefore, it would be very interesting to increase the number of sets, at least to 3, and even to gather data over a training period, of at least 4–6 weeks. Performing a new study considering and overcoming these limitations would allow us to affirm with a higher degree of scientific soundness whether the inter-set foam rolling procedure could be useful in increasing repetitions without affecting fatigue or not.

A second kind of limitation consists of the choice of the foam rolling procedure. This investigation compared foam rolling with a control condition (no intervention and passive rest). According to recent literature, it would be very interesting to replicate the study using real foam rolling, sham rolling, and a control or even agonistic static stretching. Increasing the variables would allow a better understanding of the possible relationship between foam rolling and performance, as well as its usefulness and under which conditions. Furthermore, the increase in variables may mitigate the potential placebo/nocebo effects that might be present in this study, given that the participants were unblinded to the two interventions.

Another limitation to highlight, in order to improve future investigations on this aspect, is the absence of a surface EMG measurement of the latissimus dorsi’s electrical activity. It would be very useful to understand whether foam rolling procedures modify the muscular activity during the subsequent set-in case of repetitions improvement. Data about the range of motion modification, before and after foam rolling, are missing in the current research as well. Obtaining information about the effects of the combination of strength training and foam rolling on both acute and chronic range of motion modifications could be very interesting to completely describe the experimentation.

Finally, we suggest that the findings should be corroborated through replication studies utilizing larger sample sizes to allow more robust statistical analysis. 

The authors are aware of all these limitations; for this reason, the paper must be considered merely an initial investigation aiming to open a new and interesting perspective on the upper limbs’ responses to foam rolling procedures.

## 5. Conclusions

In summary, it can be concluded that, during training with the lat pull-down exercise, the use of inter-set foam rolling procedures on the latissimus dorsi allows for an acute increase in the number of repetitions and average power without affecting fatigue immediately after the foam rolling intervention. 

## Figures and Tables

**Figure 1 jfmk-09-00043-f001:**
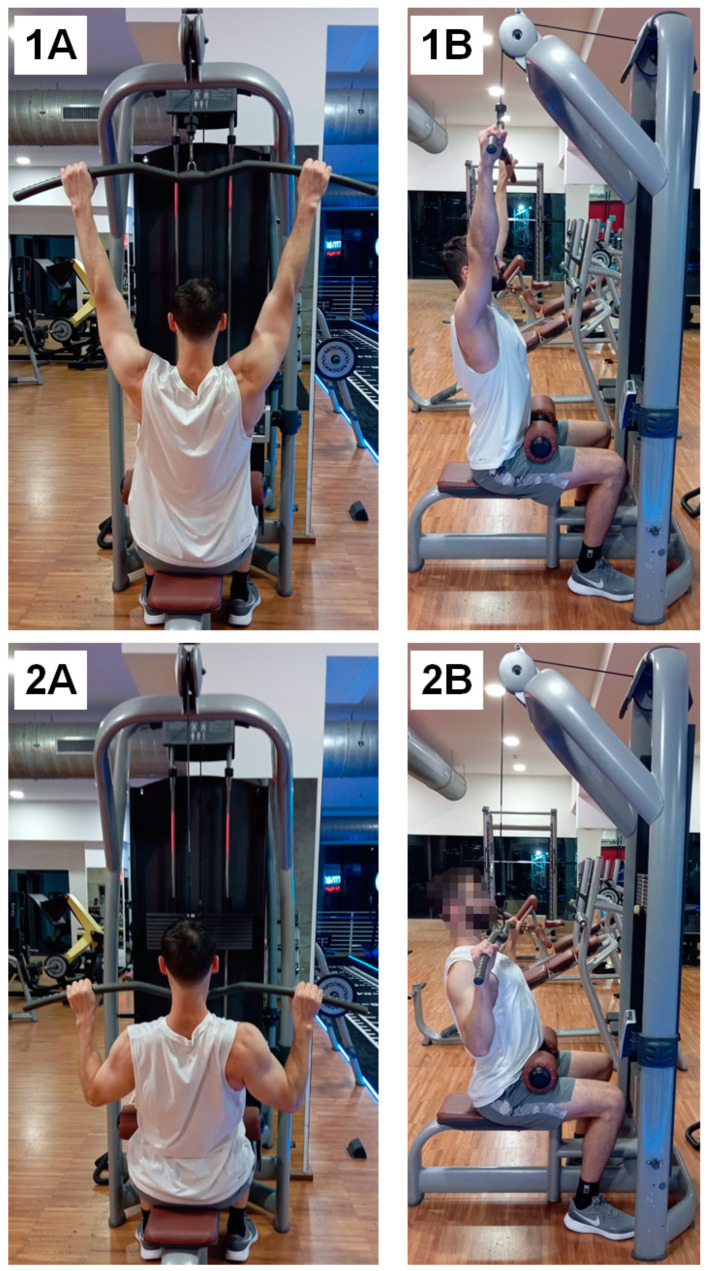
Execution of LPDE illustrating the starting (**1**) and ending (**2**) positions from the posterior (**A**) and lateral (**B**) views, respectively. During each repetition, participants were instructed to control scapular adduction and depression during the concentric phase and vice versa during the eccentric phase. Participants were instructed to maintain an upward chest and draw their shoulders backward, with elbows slightly forward in relation to the vertical axis of the trunk. In the starting position, the elbows were to be fully extended, and the concentric phase was completed by pulling the bar under the chin. Trunk oscillation was strictly prohibited.

**Figure 2 jfmk-09-00043-f002:**
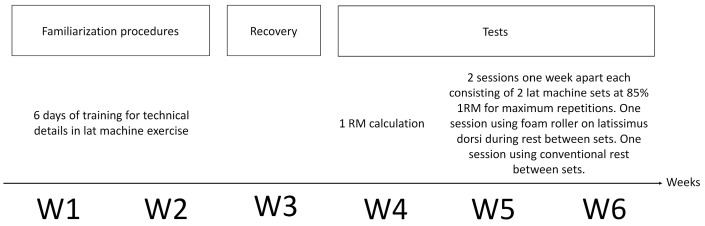
Study timeline and intervention description.

**Figure 3 jfmk-09-00043-f003:**
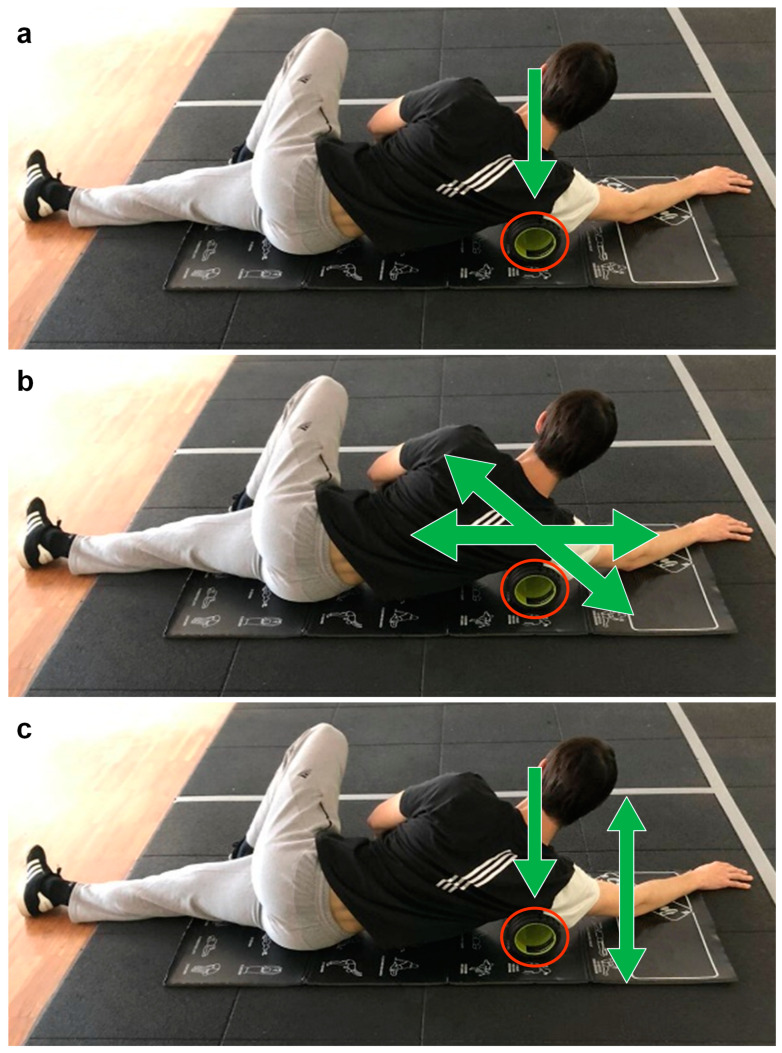
Foam rolling exercise during recovery. In the red circle the foam roller. (**a**)—first bout technique, pressing the trigger point. (**b**)—second bout technique, rolling in different directions around the trigger point. (**c**)—third bout technique, pressing the trigger point and moving the arm.

**Figure 4 jfmk-09-00043-f004:**
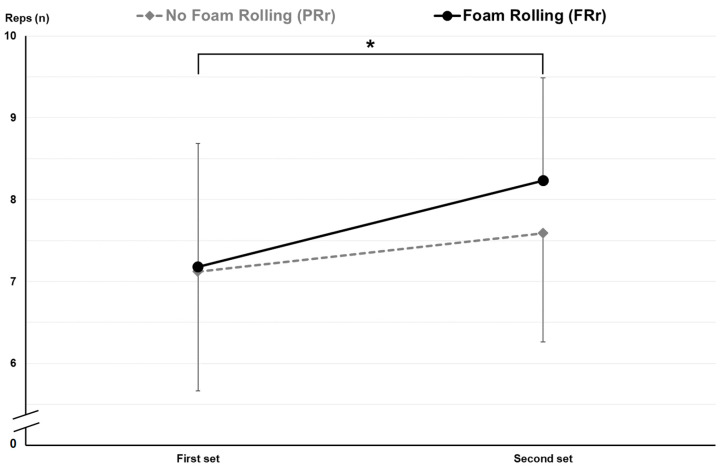
REPS changes between the first and second set in Pr and FRr. *: significant difference for FRr.

**Figure 5 jfmk-09-00043-f005:**
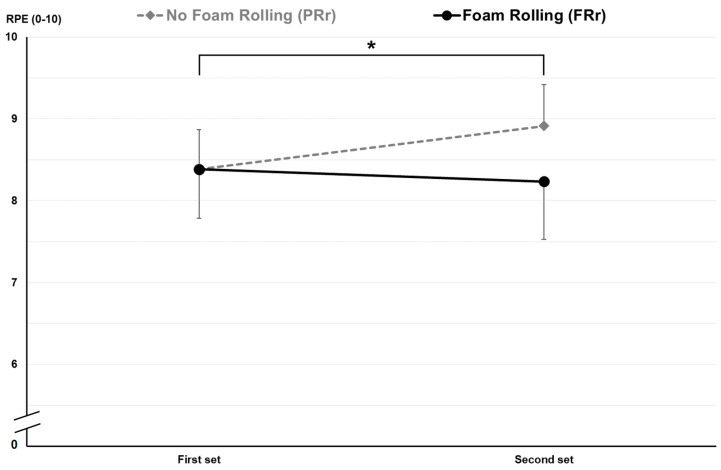
RPE changes between first and second set in Pr and FRr. *: significant difference for PR.

**Table 1 jfmk-09-00043-t001:** General and specific warm-up procedures before the 1RM calculation.

General Warm-Up	Specific Warm-Up
20 movements of each exercise: Scapula adduction/abduction Scapula elevation/depression Shoulder circumduction Shoulder flexion/extension	LPDE: 3 sets of 8-8-6 repetitions 60″ of rest between each set Initial load 50% of body mass, increasing 5 kg at the end of each set

**Table 2 jfmk-09-00043-t002:** General and specific warm-up procedures before LPDE training sessions.

General Warm-Up	Specific Warm-Up
20 movements of each exercise: Scapula adduction/abduction Scapula elevation/depression Shoulder circumduction Shoulder flexion/extension	LPDE: 1 set of 8 repetitions at 55% 1RM 60″ of rest 1 set of 8 repetitions at 60% 1RM 60″ of rest 1 set of 6 repetitions at 65% 1 RM 90″ of rest

**Table 3 jfmk-09-00043-t003:** Results of the Mann–Whitney test for the first LPDE set in both when a passive rest (Pr) and a foam rolling rest (FRr) were performed.

Variables	Pr	FRr	*p* Value
REPS (n)	7.1 ± 1.5	7.2 ± 1.5	0.860
EXC (mm)	582.1 ± 62.3	585.3 ± 64.6	0.986
AP (W)	294.7 ± 71.2	304.6 ± 61.5	0.744
RPE (a.u.)	8.4 ± 0.5	8.4 ± 0.6	0.877

Note: REPS—number of repetitions. EXC—average excursion per repetition in millimeters. AP—average power of the set in watts. RPE—rating of perceived exertion. Pr—passive recovery. FRr—foam rolling recovery.

**Table 4 jfmk-09-00043-t004:** Results of the Wilcoxon test for the comparison of the two sets of LPDE in Pr and FRr.

Variables	Pr		*p*	FRr		*p*
	Set 1	Set 2		Set 1	Set 2	
REPS (n)	7.1 ± 1.5	7.6 ± 1.3	0.074	7.2 ± 1.5	8.2 ± 1.3	0.003 *
EXC (mm)	582.1 ± 62.3	584.1 ± 63.9	0.653	585.3 ± 64.6	587.5 ± 73.4	0.463
AP (W)	294.7 ± 71.2	299.7 ± 65.4	0.381	304.6 ± 61.5	318.8 ± 60.8	0.035 *
RPE (a.u.)	8.4 ± 0.5	8.9 ± 0.5	0.007 *	8.4 ± 0.6	8.2 ± 0.7	0.417

Note: REPS—number of repetitions. EXC—average excursion per repetition in millimeters. AP—average power of the set in watts. RPE—rating of perceived exertion. Pr—passive recovery. FRr—foam rolling recovery. * Significant difference.

## Data Availability

The data that support the findings of this study are available from the corresponding author, upon reasonable request.

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
