# Peer review of "Inter-Set Foam Rolling of the Latissimus Dorsi Acutely Increases Repetitions in Lat Pull-Down Exercise without Affecting RPE"

_jfmk, 2024, doi:10.3390/jfmk9010043_

Round 1
Reviewer 1 Report
Comments and Suggestions for Authors
The study aimed to assess the effect of foam rolling as a recovery intervention between exercise sets. The novelty of the research comes from the exercise and musculature under study (lat pull-down and latissimus dorsi, respectively). The manuscript is well written and the data collection appears to be robust, however I have concerns over the approach taken to analyse the data, and therefore the validity of conclusions that can be drawn.
Specific comments:
It is unclear why the study is referred to as a pilot study, the authors report an a priori sample size calculation and a hypothesis to be tested, these are both features that do not exist in pilot studies. (I advise the authors to refer to "https://www.nccih.nih.gov/grants/pilot-studies-common-uses-and-misuses" for clarification).
Introduction
The mechanism described in L47-49 to explain foam rolling (and the term "myfoscial release" that is used), are not supported by existing evidence. This point is actually discussed in the article the authors have cited (Behm, D. G., & Wilke, J. (2019). Do self-myofascial release devices release myofascia? Rolling mechanisms: a narrative review. Sports Medicine, 49(8), 1173-1181.).
To strengthen the rationale for the study, authors should: 1) explain why the rest period between sets was the target for foam rolling treatment (e.g., as opposed to pre-exercise) [L82], and 2) justify why existing evidence from other muscle groups cannot be applied to this situation (i.e., what about latissimus dorsi and lat pull-down exercise means that results might be different from those observed when other muscles are treated with foam rolling/ other exercises are performed?).
The basis for the hypothesis is unclear. No evidence has been presented in the introduction to suggest that foam rolling leads to over-stimulation of targeted muscles [L84-85], and evidence has been cited that demonstrates a lack of negative effects associated with functional performance following foam rolling [52-55].
Materials and Methods
As stated above, it is unclear why a sample size calculation would be included in a pilot study, but since it appears that "pilot study" is an unsuitable classification for this research, the sample size should be retained, but more detail is required on this (including the software used to estimate sample size, and the source of the data used in the calculation).
The description presented in L156-159 is confusing, it is not clear what time period(s) in the study are being referred to here.
It is unclear why non-parametric testing should be adopted [L233-234] when a sample size calculation has been reported to indicate that the study was sufficiently powered, and a normal distribution was observed.
Data have been pooled within each condition for comparisons of treatment effect and then separately analysed for comparisons of time (i.e., set) effect. This analysis is inappropriate, since multiple comparisons have been performed (please refer to "Ranganathan, P., Pramesh, C. S., & Buyse, M. (2016). Common pitfalls in statistical analysis: the perils of multiple testing. Perspectives in clinical research, 7(2), 106."); this results in a high chance of type I error. Since the results section, discussion, and conclusions are presently based on these analyses with a high probability of type I error, the analysis should be amended before a meaningful review of the subsequent sections can be provided.
Author Response
Dear Reviewer 1,
We would like to thank you for the time allowed to this review process. As a result, we are submitting the revised version for a possible publication in this respectable Journal. Below, you can find our responses; each comment is followed by its respective reply. We made changes in the manuscript in order to address suggestions and make it clearer for the readers. Our responses in the manuscript appear using the track change instrument. We very much appreciate your comments on the document, which have helped us to improve its quality.
All authors have made sufficient contributions, responded to your comments and have approved the submitted manuscript.
Best regards,
The Authors
Legend:
R1(Reviewer 1)
A (Authors)
1) R1:
The study aimed to assess the effect of foam rolling as a recovery intervention between exercise sets. The novelty of the research comes from the exercise and musculature under study (lat pull-down and latissimus dorsi, respectively). The manuscript is well written and the data collection appears to be robust, however I have concerns over the approach taken to analyse the data, and therefore the validity of conclusions that can be drawn.
A: We thank the Reviewer 1 for appreciating our work and we provide some modifications in the following lines to improve the paper.
2) R1:
Specific comments:
It is unclear why the study is referred to as a pilot study, the authors report an a priori sample size calculation and a hypothesis to be tested, these are both features that do not exist in pilot studies. (I advise the authors to refer to "https://www.nccih.nih.gov/grants/pilot-studies-common-uses-and-misuses" for clarification).
A: We thank the Reviewer 1 for underlying this important methodological aspect. We removed from our text the term “pilot study”.
3) R1:
Introduction
The mechanism described in L47-49 to explain foam rolling (and the term "myfoscial release" that is used), are not supported by existing evidence. This point is actually discussed in the article the authors have cited (Behm, D. G., & Wilke, J. (2019). Do self-myofascial release devices release myofascia? Rolling mechanisms: a narrative review. Sports Medicine, 49(8), 1173-1181.).
A: We remodulate the sentence using the same words written in the “Key Points” box of the cited paper.
4) R1:
To strengthen the rationale for the study, authors should: 1) explain why the rest period between sets was the target for foam rolling treatment (e.g., as opposed to pre-exercise) [L82], and 2) justify why existing evidence from other muscle groups cannot be applied to this situation (i.e., what about latissimus dorsi and lat pull-down exercise means that results might be different from those observed when other muscles are treated with foam rolling/ other exercises are performed?).
A: In the discussion we gave a detailed explanation of the reason why we performed the study (lines 308-318). Following the suggestion of the Reviewer 1 we modified the introduction in order to give more strength to the rationale of the study, lines 85-91.
5) R1:
The basis for the hypothesis is unclear. No evidence has been presented in the introduction to suggest that foam rolling leads to over-stimulation of targeted muscles [L84-85], and evidence has been cited that demonstrates a lack of negative effects associated with functional performance following foam rolling [52-55].
A: As we wrote in the previous comment, we modified the final part of the introduction, citing two researches about the effect of inter-set foam rolling. These papers demonstrate negative effect on the subsequent set. Since the first version of the paper we were referring to those papers but maybe we lost in our management of the main text. Now we are sure that the doubts of the Reviewer 1 are solved.
6) R1:
Materials and Methods
As stated above, it is unclear why a sample size calculation would be included in a pilot study, but since it appears that "pilot study" is an unsuitable classification for this research, the sample size should be retained, but more detail is required on this (including the software used to estimate sample size, and the source of the data used in the calculation).
A: Done, line 111.
7) R1:
The description presented in L156-159 is confusing, it is not clear what time period(s) in the study are being referred to here.
A: We moved the part you mention in a separated paragraph to be clearer (Paragraph 2.3.2 lines 181-189). Moreover, we underline the week of test referring to the Figure 2. We thank the Reviewer 1 for the suggestion because now the paper is clearer.
8) R1:
It is unclear why non-parametric testing should be adopted [L233-234] when a sample size calculation has been reported to indicate that the study was sufficiently powered, and a normal distribution was observed.
A: It is worth noting that the Shapiro–Wilk test showed a normal distribution for the “most of the data” and not for all the data (line 262-263). Moreover the sample size was only 17 subjects. Due to these two important aspects (lack of normal distribution of all data and poor numerousness of the sample), our statistical expert suggest to use non-parametric procedures. We also made parametric tests in our previous analysis and the results were the same but we prefer to report on the paper non-parametric results for the above mentioned reasons.
9) R1:
Data have been pooled within each condition for comparisons of treatment effect and then separately analysed for comparisons of time (i.e., set) effect. This analysis is inappropriate, since multiple comparisons have been performed (please refer to "Ranganathan, P., Pramesh, C. S., & Buyse, M. (2016). Common pitfalls in statistical analysis: the perils of multiple testing. Perspectives in clinical research, 7(2), 106."); this results in a high chance of type I error. Since the results section, discussion, and conclusions are presently based on these analyses with a high probability of type I error, the analysis should be amended before a meaningful review of the subsequent sections can be provided.
A: We thank the Reviewer 1 for this suggestion and for the reference. We think that there is a misunderstanding. We never pooled the data “pooled within each condition for comparisons of treatment effect and then separately analysed for comparisons of time (i.e., set) effect”. We just looked for the baseline performance of set n.1 between the two tests day, in order to be sure that the baseline was identical. We did not pool the data.
Then, we did not look for comparison between groups about the Pr or FRr effect. We just look for one comparison over the time (Set 1 and Set 2) in the two separated resting modalities. The reference you suggested us is explicit “In any study, when two or more groups are compared, there is always a chance of finding a difference between them just by chance. This is known as a Type 1 error” we did not compare the group over the time and we did not analyze the data multiple times but just one time.
Our statistical expert suggested us this choice. If the Reviewer 1 thinks we can write better the text to highlight this aspect we can improve the clarity of the text.
Reviewer 2 Report
Comments and Suggestions for Authors
Thank you for allowing me to review your paper. I enjoyed reading it and appreciate the work you put into it.
A few odd grammatical choices should be addressed.
Line 20 “diffused” is used correctly by its definition, but not commonly used in this way. Something simple like “used” would work fine here.
Line 67 “state of art” perhaps something like “application” or “practice” makes a little more sense here.
Line 97 “since 2 years at least” should be changed. Easiest change would probably be “for at least 2 years.”
I understand the journals want to know why a study is novel. However, I feel that the added sentence on lines 86-88 could be shortened and included previously, maybe around line 66 “making this study a novelty.” As it stands, I don’t feel the sentence adds much in what it says or where it is placed.
Line 105 what do you mean by the testing environment was controlled? Room temperature, time of day, verbal encouragement?
I think that using the encoder was a clever way to ensure full ROM was used. On line 118 the grammar is a bit strange. Easy fix would be “A common mistake when performing LPDE is reduction of ROM due to fatigue during repetitions.”
Line 120 I have a similar comment as line 20 where “constancy” is fine by definition but “consistency” is probably the more commonly used English.
Line 145 change to “testing started”
Line 161-163: To be clear they could not perform any rowing or other vertical pulling exercises (e.g. pull ups) during this time but they could perform lat pull downs?
Line 175: “based”
Line 195: You mean the intervention was different, not the rest time?
Line 203: was music playing in the gym at the time?
Comments on the Quality of English LanguageAll comments are included in previous section.
Author Response
Dear Reviewer 2,
We would like to thank you for the time allowed to this review process. As a result, we are submitting the revised version for a possible publication in this respectable Journal. Below, you can find our responses; each comment is followed by its respective reply. We made changes in the manuscript in order to address suggestions and make it clearer for the readers. Our responses in the manuscript appear using the track change instrument. We very much appreciate your comments on the document, which have helped us to improve its quality.
All authors have made sufficient contributions, responded to your comments and have approved the submitted manuscript.
Best regards,
The Authors
Legend:
R2 (Reviewer 2)
A (Authors)
1) R2:
Thank you for allowing me to review your paper. I enjoyed reading it and appreciate the work you put into it.
A few odd grammatical choices should be addressed.
A: We thank the Reviewer 2 in appreciating our work. We will provide proper answers in the following lines. We think that the language corrections suggested by the Reviewer can improve the paper quality.
2) R2:
Line 20 “diffused” is used correctly by its definition, but not commonly used in this way. Something simple like “used” would work fine here.
A: Done.
3) R2:
Line 67 “state of art” perhaps something like “application” or “practice” makes a little more sense here.
A: Done.
4) R2:
Line 97 “since 2 years at least” should be changed. Easiest change would probably be “for at least 2 years.”
A: Done.
5) R2:
I understand the journals want to know why a study is novel. However, I feel that the added sentence on lines 86-88 could be shortened and included previously, maybe around line 66 “making this study a novelty.” As it stands, I don’t feel the sentence adds much in what it says or where it is placed.
A: Done, line 70.
6) R2:
Line 105 what do you mean by the testing environment was controlled? Room temperature, time of day, verbal encouragement?
A: We mean that no risk for the participants were present. In the 2.3 paragraph we reported “a fitness gym center at a mean temperature of 20 °C and a mean relative humidity of 51%, and each subject was tested at the same time of the day, avoiding any circadian effect [24-26]”.
7) R2:
I think that using the encoder was a clever way to ensure full ROM was used. On line 118 the grammar is a bit strange. Easy fix would be “A common mistake when performing LPDE is reduction of ROM due to fatigue during repetitions.”
A: Done.
8) R2:
Line 120 I have a similar comment as line 20 where “constancy” is fine by definition but “consistency” is probably the more commonly used English.
A: Done.
9) R2:
Line 145 change to “testing started”
A: Done.
10) R2:
Line 161-163: To be clear they could not perform any rowing or other vertical pulling exercises (e.g. pull ups) during this time but they could perform lat pull downs?
A: Exactly yes. During the testing period only LPDE on days of test was allowed. We add a specification in lines 186-187.
11) R2:
Line 175: “based”
A: Done.
12) R2:
Line 195: You mean the intervention was different, not the rest time?
A: No, we mean that the two sessions were identical but the rest modalities were different. We modified the text in order to be clearer, line 222-223.
13) R2:
Line 203: was music playing in the gym at the time?
A: Thanks for this comment because the Reviewer 2 remember us to add in the paper an important aspect that we took in consideration. Indeed, to avoid influence of different music traces we shut down the speaker in that part of the gym during the testing procedures. We add this information in the paper, line 233-2355
Round 2
Reviewer 1 Report
Comments and Suggestions for Authors
I thank the authors for their diligent revision of the manuscript. The Introduction and Methods sections have both been improved by the work carried out since the previous review.
Regarding the statistical approach adopted; taking the authors' responses into account, I would still advise that performing a Kruskal–Wallis test with Dunn post hoc comparisons would be the more suitable approach, rather than Mann-Whitney comparison of treatments and Wilcoxon comparison of time/set. However, if the authors are satisfied with their current analysis, this need not be revised. On the other hand, it would be appropriate to include a further statement in the limitations section to acknowledge that the findings should be corroborated through replication studies utilizing larger sample sizes to allow more robust statistical analysis.
Author Response
thanks for any suggestion
We updated the manuscript in "study Limitation section"
Finally, We suggest that the findings should be corroborated through replication studies utilizing larger sample sizes to allow more robust statistical analysis.